# Unleashing the Autoconversion Rates Forecasting: Evidential Regression from Satellite Data

**Maria Carolina Novitasari[1]**
maria.novitasari.20@ucl.ac.uk

**Johannes Quaas[2,3]**
johannes.quaas@uni-leipzig.de

**Miguel R. D. Rodrigues[1]**
m.rodrigues@ucl.ac.uk

[1]Department of Electronic and Electrical Engineering, University College London
[2]Leipzig Institute for Meteorology, Universität Leipzig
[3]ScaDS.AI - Center for Scalable Data Analytics and AI, Universität Leipzig

## Abstract

High-resolution simulations such as the ICOsahedral Non-hydrostatic Large-Eddy Model (ICON-LEM) can be used to understand the interactions between aerosols, clouds, and precipitation processes that currently represent the largest source of uncertainty involved in determining the radiative forcing of climate change. Nevertheless, due to the exceptionally high computing cost required, this simulation-based approach can only be employed for a short period of time within a limited area. Despite the fact that machine learning can mitigate this problem, the related model uncertainties may make it less reliable. To address this, we developed a neural network (NN) model powered with evidential learning to assess the data and model uncertainties applied to satellite observation data. Our study focuses on estimating the rate at which small droplets (cloud droplets) collide and coalesce to become larger droplets (raindrops) – autoconversion rates – since this is one of the key processes in the precipitation formation of liquid clouds, hence crucial to better understanding cloud responses to anthropogenic aerosols. The results of estimating the autoconversion rates demonstrate that the model performs reasonably well, with the inclusion of both aleatoric and epistemic uncertainty estimation, which improves the credibility of the model and provides useful insights for future improvement.

## 1 Introduction

Future climate projections demand a deeper understanding of interactions between aerosols and clouds since they constitute the biggest uncertainty in estimating the radiative forcing of climate change [IPCC, 2021]. One way to reduce such uncertainties is to understand the interaction between aerosols, clouds, and precipitation processes. One of the high-resolution simulations that can be used to study these interactions is the ICON-LEM [Zängl et al., 2015, Dipankar et al., 2015, Heinze et al., 2017]. Nevertheless, due to the exceptionally high computing cost required, it can only be employed for a limited period and geographical area.

While machine learning has the potential to address the challenges mentioned, it is important to note that predictions made by these models may be affected by noise and model inference error [Malinin, 2019]. The degree of uncertainty in a model depends on the machine learning algorithm and the availability of data. These model uncertainties might be more pronounced for climate forecasts, raising concerns about the model's reliability.

NeurIPS 2023 AI for Science Workshop.

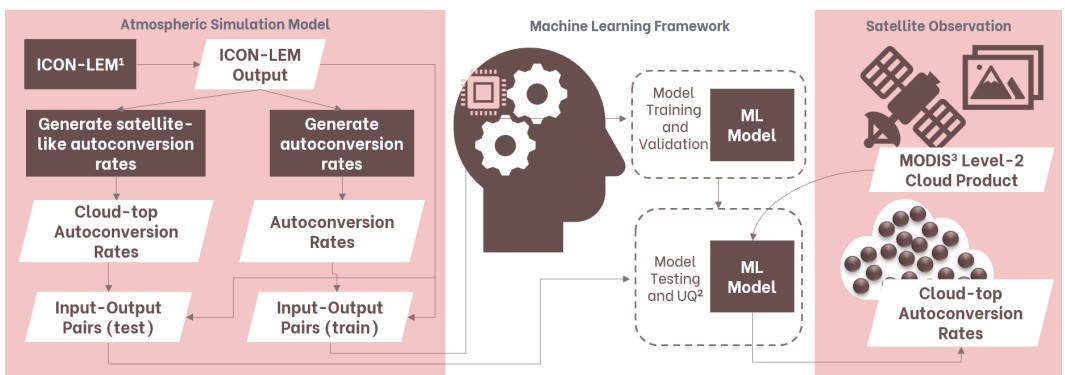

Figure 1: General framework.

The aforementioned issues are addressed in this study by employing deep evidential regression [Amini et al., 2020, Meinert et al., 2023] and the massive collection of satellite data, providing long-term global spatial coverage up to several decades. We focus in particular on autoconversion rates estimation since this is one of the key processes in the precipitation formation of liquid clouds, hence crucial to better understanding cloud responses to anthropogenic aerosols and, ultimately, climate change.

Our current work represents a continuation and enhancement of our previous research efforts. Our previous work [Novitasari et al., 2021] also considered the prediction of autoconversion rates from satellite data; however, it entailed a more complex approach involving a machine learning model with more features and higher number of trainable weights, resulting in increased computational costs compared to our work. Furthermore, our previous approach utilized a more complex simulator (COSP; Swales et al., 2018) to generate satellite-like data, while our current approach is more cost-effective and straightforward as we simply define the cloud-top information based on cloud optical thickness. Importantly, we have not quantitatively assess the inherent uncertainty in our previous findings. This work offers a computationally effective solution by employing the fewest features necessary and minimizing trainable weights while still obtaining accurate predictions of autoconversion rates. Additionally, deep evidential learning evaluates data and model uncertainties without the need for additional training or inference.

## 2 Proposed Approach

We present a novel approach for direct autoconversion rate extraction from satellite observation, which includes the estimation of data and model uncertainties. The general framework, as illustrated in Figure 1, involves climate science-based steps for generating training and testing datasets, along with a machine learning framework incorporating uncertainty quantification methods, as elaborated in the following section.

**Datasets** We use datasets from ICON-LEM output over Germany on 2 May 2013, where distinct cloud regimes occurred, allowing for the investigation of quite different elements of cloud formation and evolution [Heinze et al., 2017]. We study a time period of 09:55 UTC to 13:20 UTC. Our focus is on ICON-LEM with a 156 m resolution on the native ICON grid, then regridded to a regular 1 km resolution to match the resolution of MODIS. The autoconversion rates in our training and testing data were derived using the two-moment microphysical parameterization of Seifert and Beheng (2006). The autoconversion rates for cloud tops that simulate satellite data were determined by selecting rates where the cloud optical thickness (COT), calculated from top to bottom, exceeds 1. The optical thickness represents the extent to which optical satellite sensors can retrieve cloud microphysical information.

We use dataset of ICON numerical weather prediction (ICON-NWP) Holuhraun which were performed over Holuhraun volcano for a week from 1 to 7 September 2014 to further test the performance of our machine learning models [Kolzenburg et al., 2017, Haghighatnasab et al., 2022]. The dataset has a horizontal resolution of approximately 2.5 km. As for the satellite observation data displayed

on the right side of Figure 1, we use cloud product level-2 of Terra and Aqua MODIS [Platnick et al., 2017, 2018].

**Uncertainty Quantification Methods** To capture the uncertainty associated with the predicted autoconversion rates, we employ the evidential regression framework proposed by Amini et al. (2020). However, we incorporate the modified version proposed by Meinert et al. (2023) because the regularizer formulation proposed by Amini et al. (2020) is insufficient for finding the marginal likelihood parameters.

Amini et al. (2020) approach the regression problem by representing it as a normal distribution with an unknown mean and variance, $N(\mu, \sigma^2)$. They adopt a normal prior distribution for the mean, with $\mu$ following $N(\gamma, \sigma^2 \nu^{-1})$, and an inverse Gamma prior distribution for the variance, with $\sigma^2$ following $\Gamma^{-1}(\alpha, \beta)$. As a result, they obtain a combined prior distribution known as Normal Inverse-Gamma, characterized by the parameters $\gamma, \nu, \alpha$, and $\beta$.

Given a set of $(x_i, y_i)$ pairs, the overall loss function $\mathcal{L}_i(w)$ for the NN, as defined by Amini et al.(2020), where $w$ represents a set of weights, is formulated as: $\mathcal{L}_i(w) = \mathcal{L}_i^{\text{NLL}}(w) + \lambda \mathcal{L}_i^{\text{R}}(w)$, where $\mathcal{L}_i^{\text{NLL}}$ represents the negative log-likelihood to maximize the model fit, defined as:

$$\mathcal{L}_i^{\text{NLL}}(w) = \frac{1}{2} \log\left(\frac{\pi}{\nu}\right) - \alpha \log \Omega + \left(\alpha + \frac{1}{2}\right) \log\left((y_i - \gamma)^2 \nu + \Omega\right) + \log\left(\frac{\Gamma(\alpha)}{\Gamma\left(\alpha + \frac{1}{2}\right)}\right) \quad (1)$$

where $\Omega = 2\beta(1 + \nu)$. $\mathcal{L}_i^{\text{R}}$ denotes the regularization term, which is scaled by a regularization coefficient, $\lambda$. Choosing a low value for $\lambda$ could lead to overconfidence, whereas opting for a high value might result in excessive uncertainty. However, instead of utilizing the L1 error, we employ an adjustment to the residuals by incorporating the width of the student-t distribution ($w_{St}$), as recommended by Meinert et al. (2023).

Consequently, the regularization term can be expressed as:

$$\mathcal{L}_i^{\text{R}}(w) = \left|\frac{y_i - \gamma}{w_{St}}\right|^p \Phi \quad (2)$$

where $w_{St}$ is computed as $\sqrt{\frac{\beta(1+\nu)}{\alpha \cdot \nu}}$. $w_{St}$ is also employed for evaluating the aleatoric uncertainty. Conversely, the epistemic uncertainty is defined as $\frac{1}{\sqrt{\nu}}$. The total evidence, denoted as $\Phi$, is defined as $\nu + 2\alpha$ [Meinert et al., 2023]. Additionally, the parameter $p$ serves as an additional hyperparameter, determining the magnitude of residual impact within regularization.

**Machine Learning Models** In this study, we trained our machine learning models using ICON-LEM output as input. The input to our machine learning model is the cloud effective radius as this is one of the crucial parameters of the cloud microphysical state that is typically obtained from satellite retrievals [Platnick et al., 2017, Grosvenor et al., 2018]. The machine learning model output, serving as groundtruth, is the autoconversion rates derived from ICON-LEM. The preprocessing step is detailed in Appendix A.

We then trained and tested a deep evidential regression [Amini et al., 2020, Meinert et al., 2023] model with an evidential regularizer coefficients, $\lambda$, set to 1e-6, chosen for optimal calibration (details in Appendix C). Additionally, following Meinert et al. [2023], the hyperparameter $p$ was chosen with the value of 2. The shallow NN is trained to infer the hyperparameters of the evidential distribution by applying evidential priors over the original Gaussian likelihood function.

In the early stages of our testing process, we compared our models' performance against simulation data. We then proceed to calculate the data (aleatoric) and model (epistemic) uncertainties, rigorously evaluating their calibration through the implementation of the spread-skill plot [Wilks, 2011, Haynes et al., 2023] and discard test [Barnes and Barnes, 2021, Haynes et al., 2023] methods. Finally, we employ our shallow NN model to make direct predictions of the autoconversion rates from satellite data, including the estimation of its data and model uncertainties.

We selected a shallow NN model for its superior performance compared to other machine learning models as shown in Appendix B. Despite the fact that DNN model produced the best outcome in

Table 1: Evaluation of autoconversion rates prediction results on simulation data (ICON) using NN Evidential model.

| Data | $R^2$ | MAPE | RMSPE | SSIM | PSNR |
|---|---|---|---|---|---|
| (a) ICON-LEM Germany | 90.54% | 9.14% | 11.27% | 90.11% | 26.30 |
| (b) Cloud-top ICON-LEM Germany | 89.88% | 10.86% | 13.89% | 89.96% | 25.89 |
| (c) Cloud-top ICON-NWP Holuhraun | 84.39% | 8.56% | 10.73% | 91.50% | 25.81 |

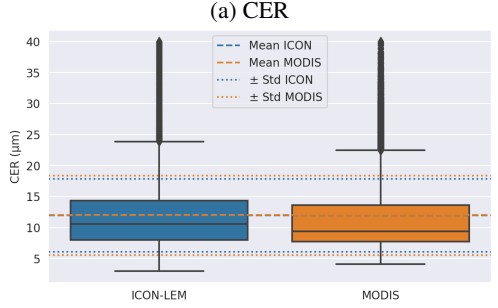 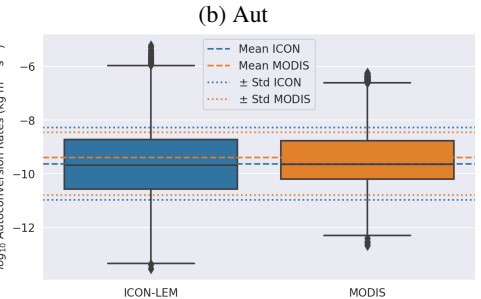

Figure 2: Mean, standard deviation (Std), median, and percentiles (p25, p75) of cloud-top ICON-LEM and MODIS variables over Germany: cloud effective radius (CER) and autoconversion rates (Aut).

our scenario (see Appendix B), we deem the NN model to be superior due to its ability to produce comparable outcomes with significantly fewer trainable weights. Additionally, the use of NNs (including DNN) can be coupled with evidential learning to enable estimation of aleatoric and epistemic uncertainties without additional training.

## 3  Experimental Results

**Autoconversion on Simulation Data (ICON)**    This stage involves testing our shallow NN model in 3 scenarios with different datasets: (1) ICON-LEM Germany (different times), (2) Cloud-top ICON-LEM Germany (satellite-like data), and (3) Cloud-top ICON-NWP Holuhraun (different data, time, location, and resolution), details in Appendix E.1. All findings, displayed in Table 1, exhibit strong performance with SSIM around 90%. This confirms our approach's ability to accurately estimate autoconversion rates using simulation data. Visual representations are available in Appendix E.2. The details of the model architecture and additional experimental results with a different NN (DNN) architecture from our previous work [Novitasari et al., 2021] are also available in Appendix D.

**Autoconversion on Satellite Observation (MODIS)**    This stage involves testing our models on satellite data, MODIS Aqua over Germany (latitude: 47.50° to 54.50° N, longitude: 5.87° to 10.00° E) on 2 May 2013 at 13:20 local time. Even though satellite predictions cannot be directly compared with simulation results – due to the fact that the ICON-LEM simulation does not put clouds in their exact right places – the autoconversion rates obtained from the simulation output and the predicted autoconversion rates from satellite data demonstrated statistical concordance, as shown in Figure 2. The mean, standard deviation, median, 25th and 75th percentiles of the autoconversion rates of the cloud-top ICON-LEM Germany compared to MODIS are relatively close. Additional results in Appendix E.3. This implies that our approach is capable of estimating autoconversion rates directly from satellite data.

**Evaluation of Uncertainty Estimation**    The total uncertainty (aleatoric and epistemic) performance of the deep evidential regression on simulation data was evaluated using the spread-skill plot and discard test. Results showed that the deep evidential regression with a coefficient $\lambda$ of 1e-6 was well-calibrated, as depicted in Figure 3. This is demonstrated by spread skill plots that are very close to the ideal line and discard test that decrease monotonically from left to right.

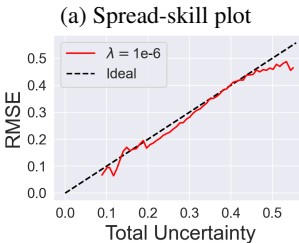
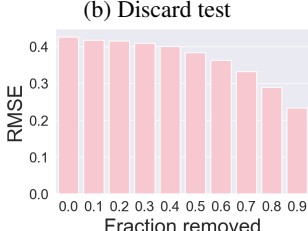

Figure 3: Evaluation of uncertainty estimation on simulation data (ICON).

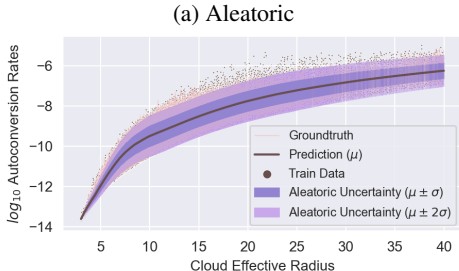
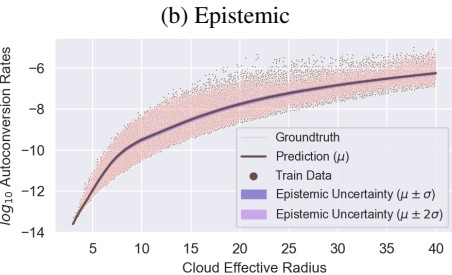

Figure 4: (a) Aleatoric and (b) epistemic uncertainty estimates of autoconversion rates $(\mathrm{kg\,m^{-3}\,s^{-1}})$ prediction using atmospheric simulation data (ICON) over Germany.

**Uncertainty Estimation on Satellite-like and Satellite Data**    Figures 4 displays the visualization of aleatoric and epistemic uncertainties on satellite-like data, over Germany (see Appendix E.4 for satellite data over Germany, as well as satellite-like and satellite data over Holuhraun). The plots reveal a wider range of aleatoric uncertainty compared to epistemic uncertainty. This observation aligns with the fact that a single independent variable can correspond to multiple ranges of dependent variables, as shown in Figure 4.

## 4    Conclusion

In this study, we provide a computationally efficient solution to unravel the key process of precipitation formation for liquid clouds, the autoconversion process, from satellite data by employing the fewest attributes necessary while still obtaining meaningful results. A thorough investigation, with details provided in the appendix, is also included. Furthermore, we employ evidential learning to estimate data and model uncertainties, eliminating the need for additional training or inference, hence reducing the overall costs. It has demonstrated good calibration, enhancing the model's credibility, and providing valuable insights for future improvements. Our findings show that data uncertainty contributes the most to the overall uncertainty, hence modifying the model architecture is unlikely to improve outcomes significantly. Instead, it would be more effective to prioritize enhancing data quality or incorporating an additional crucial feature, such as COT per layer. Unfortunately, satellite data lacks this information, but future efforts will focus on estimating this feature to enhance autoconversion rates estimation and reduce uncertainty.

## Acknowledgments and Disclosure of Funding

We would like to express our sincere appreciation to the anonymous reviewers for their valuable feedback. This research receives funding from the European Union's Horizon 2020 research and innovation programme under Marie Skłodowska-Curie grant agreement No 860100 (iMIRACLI). This work used resources of the Deutsches Klimarechenzentrum (DKRZ) granted by its Scientific Steering Committee (WLA) under project ID bb1143. The model output data used for the development of the research in the frame of this scientific article is available on request from in tape archives at the DKRZ, which will be accessible for 10 years.

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

# A Machine Learning Models

We split the data into 80% for training and validation, and 20% for testing. To enhance the model performance, logarithmic transformations are applied to both the input and output variables for normalization. This normalization process effectively addresses data presented with extremely small numerical values, thereby improving interpretability and stability. Furthermore, the input variables undergo additional normalization using standard scaling techniques.

The performance of each model is evaluated by calculating a range of metrics, including $R^2$, MAPE (Mean Absolute Percentage Error), RMSPE (Root Mean Squared Percentage Error), Peak Signal-to-Noise Ratio (PSNR), and Structural Similarity Index (SSIM), on the testing data. Prior to calculating each metric, the data is normalised by transforming it using base 10 logarithms and then scaling it to a range between 0 and 1.

# B Initial Machine Learning Models (without Evidential Learning)

Table B1: Evaluation of the autoconversion rates prediction results on simulation model using various machine learning models (linear regression, second order polynomial regression, random forest, shallow neural network, and deep neural network) - ICON-LEM Germany.

| Method | $R^2$ | MAPE | RMSPE | SSIM | PSNR (dB) |
|--------|-------|------|-------|------|-----------|
| LR | 88.77% | 10.86% | 15.28% | 88.39% | 25.56 |
| PR | 90.03% | 9.57% | 12.22% | 89.96% | 26.08 |
| RF | 90.53% | 9.15% | 11.54% | 90.09% | 26.30 |
| NN | 90.54% | 9.15% | 11.35% | 90.11% | 26.31 |
| DNN | 90.54% | 9.15% | 11.29% | 90.11% | 26.31 |

Table B1 presents the results of the initial series of experiments. It is evident from the table that a simple linear regression (LR – which is discussed in the book of Gelman and Hill [2006]) model yields fairly good results, with $R^2$ and SSIM values exceeding 88%. The performance can be further improved by switching to a second-degree polynomial regression (PR) model. Out of all the techniques tested, deep neural network (DNN – [Schmidhuber, 2015]) produces the best results with $R^2$ and SSIM values exceeding 90%. Random forest (RF – [Breiman, 2001]) also performed admirably but not superior to the DNN model with a very slight difference. The shallow neural network (NN – which was first proposed by McCulloch and Pitts [1943]) also demonstrated comparable results to the DNN model, albeit with slightly elevated RMSPE values, thereby suggesting that the DNN model still yielded the best results. Despite the fact that DNN model produced the best outcome in our scenario, we deem the NN model to be superior due to its ability to produce comparable outcomes with significantly fewer trainable weights. These findings suggest that there is a strong association between the liquid cloud particle size (cloud effective radius) and the autoconversion rates.

The DNN architecture of our machine learning model consists of five fully connected hidden layers, with 64 nodes in the first layer, 128 nodes in the second layer, 256 nodes in the third layer, 512 nodes in the fourth layer, and 1024 nodes in the fifth layer. Additionally, our NN is a basic neural network with a single hidden layer comprising 64 neurons.

For both neural network models (NN and DNN), the training was performed using Adam's optimizer. At each hidden layer, Leaky ReLU as the activation function is performed. The batch size and learning rate are set based on keras tuner algorithm [O'Malley et al., 2019]. However, it is worth noting that those architectures were chosen arbitrarily and alternative architectures can also be employed.

# C Selection of the Evidential Regularizer

Our evidential regression models were trained with varying evidential regularizer coefficients $\lambda$ ranging from 1e-2 to 1e-9. Additionally, following Meinert et al. [2023], the hyperparameter $p$ was chosen with the value of 2. The calibration of uncertainty estimation was evaluated using a spread-skill plot, and the most well-calibrated coefficient was selected. The results of this experiment is shown in Figure C1.

The best coefficient $\lambda$ was found to be 1e-6 as it was closest to the ideal line in the spread-skill plot. The ideal line represents a scenario where the predicted uncertainty matches the actual error of prediction. A value above the diagonal indicates overconfidence in the model's uncertainty estimate, while a value below the diagonal represents underconfidence.

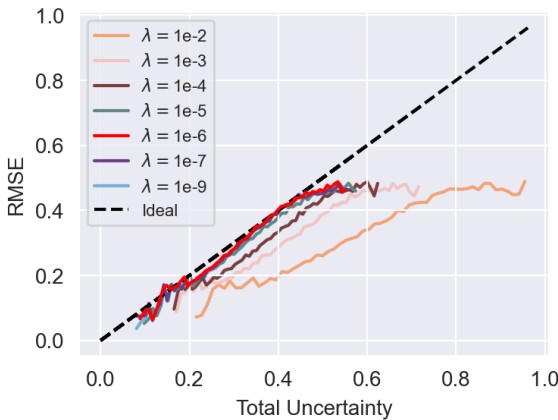

Figure C1: Spread-skill plot of deep evidential regression model (based on shallow NN model) with varying evidential regularizer coefficients.

Table D1: Evaluation of the autoconversion rate prediction results on simulation model (ICON-LEM Germany) using evidential machine learning models, including both shallow neural network (NN) and deep neural network (DNN) architectures.

| Method | $R^2$ | MAPE | RMSPE | SSIM | PSNR (dB) |
|--------|-------|------|-------|------|-----------|
| NN | 90.54% | 9.14% | 11.27% | 90.11% | 26.30 |
| DNN | 90.50% | 9.19% | 11.26% | 90.26% | 26.29 |

Table D2: Evaluation of the autoconversion rate prediction results on simulation model (Cloud-top ICON-LEM Germany) using evidential machine learning models, including both shallow neural network (NN) and deep neural network (DNN) architectures.

| Method | $R^2$ | MAPE | RMSPE | SSIM | PSNR (dB) |
|--------|-------|------|-------|------|-----------|
| NN | 89.88% | 10.86% | 13.89% | 89.96% | 25.89 |
| DNN | 89.95% | 10.74% | 13.65% | 90.18% | 25.93 |

# D  Evidential Machine Learning Models

To facilitate comparison, we train our deep evidential regression using two distinct neural network architectures. The first model, referred to as the shallow NN or NN, is our preferred choice as indicated in Appendix A. It is a basic neural network with a single hidden layer comprising 64 neurons. In contrast, the second model, known as the pyramid-shaped DNN, was employed in our previous study [Novitasari et al., 2021] to predict autoconversion rates. This model features a more intricate architecture and a greater number of trainable weights.

For both models, the activation function used at each hidden layer is Leaky ReLU. While for the loss function, we use evidential deep regression loss (refer to Section 2). Furthermore, the training was performed using Adam's optimizer. The batch size and learning rate are set based on keras tuner algorithm. Given our use of evidential deep learning, we modify the final layer of both models to incorporate the evidential component.

The details of the training and testing data utilized are described in Section 2. We use a lambda coefficient of 1e-6, determined as the most well-calibrated coefficient based on previous experiments (refer to Appendix C for more details), and additionally, following Meinert et al. [2023], the hyperparameter $p$ was chosen with the value of 2.

The comparison results, presented in Tables D1, D2 and D3, demonstrate that all models exhibit comparable performance, delivering quite good results for different testing datasets and scenarios (see Appendix E.1). Despite the fact that DNN model yielding slightly better results in our specific scenario, we consider the NN model (our current approach) to be preferable since it can achieve comparable outcomes with significantly fewer trainable weights.

Table D3: Evaluation of the autoconversion rate prediction results on simulation model (Cloud-top ICON-NWP Holuhraun) using evidential machine learning models, including both shallow neural network (NN) and deep neural network (DNN) architectures.

| Method | $R^2$ | MAPE | RMSPE | SSIM | PSNR (dB) |
|--------|-------|------|-------|------|-----------|
| NN | 84.39% | 8.56% | 10.73% | 91.50% | 25.81 |
| DNN | 84.99% | 8.39% | 10.83% | 91.77% | 25.98 |

In line with our findings regarding epistemic uncertainty (refer to Section 3), modifying the model architecture is unlikely to result in significant improvements. Alternatively, it is advisable to prioritize enhancing data quality or incorporating additional crucial features, if available.

# E    Additional Results

## E.1    Testing Dataset/Scenarios

We evaluate our NN model using different testing datasets and scenarios associated with the ICON-LEM simulations over Germany and the ICON-NWP simulations over Holuhraun, as follows:

1. *ICON-LEM Germany*: In this testing scenario, we evaluate the performance of our machine learning models using the same data that was utilised during its training process. This data, which consists of a set of cloud effective radius and autoconversion rates corresponding to different points in three-dimensional space, was collected through the use of ICON-LEM simulations specifically over Germany. The testing data, however, differs from the training data as we focus on a different time period, specifically 2 May 2013 at 1:20 pm. This approach enables us to assess the model's generalisation capability to new data within the same region and day, while considering significant weather variations that evolved considerably (Heinze et al. [2017]). Number of data points: approximately 950,000.

2. *Cloud-top ICON-LEM Germany*: In this testing scenario, we evaluate the performance of our machine learning model by utilising the same data as in the previous scenario, with the exception that we are only considering the cloud-top information of the data. The data involved in this testing scenario is a collection of cloud-top autoconversion rates and cloud-top effective radius pairs associated with 2D spatial points, representing a specific range of latitude and longitude, as well as a specific altitude for each point, specifically the cloud-top height for each point. We extract this cloud-top 2D data from the 3D atmospheric simulation model by selecting the variable value at any given latitude and longitude where the cloud optical thickness exceeds 1, integrating vertically from cloud-top. Number of data points: approximately 144,000.

3. *Cloud-top ICON-NWP Holuhraun*: This final testing scenario uses distinct data from that of previous scenarios. In particular, we use cloud-top of ICON-NWP Holuhraun data that was acquired at a different location, time, and resolution compared with the data used in the previous scenarios. The ability of the model to perform well in the presence of new data is important in many practical applications, allowing the model to make accurate predictions on unseen data, adapting to varying geographical locations, and adapting to different metereological conditions. Number of data points: approximately 1.7 million.

## E.2    Autoconversion on Simulation Data (ICON)

A visualization of the autoconversion rates predicted by shallow NN with groundtruth can be seen in Figure E1. As depicted in the figure, the effectiveness of the NN model in capturing and replicating the main characteristics of the groundtruth is evident, as demonstrated by the strong resemblance between the groundtruth and the model predictions. Moreover, it can be observed from Figure E1 that there is a minimal discrepancy between the predicted values and the actual groundtruth in both the ICON-LEM Germany dataset and the ICON-NWP Holuhraun dataset.

## E.3    Autoconversion on Satellite Observation (MODIS)

The probability density function (PDF) of the autoconversion rates shows a slight difference at its peak, with MODIS being slightly higher than cloud-top ICON-LEM Germany, as seen in Figure E2. This difference is attributed to discrepancies in the cloud effective radius or liquid cloud particle size, which is an input variable, between MODIS and cloud-top ICON-LEM over Germany.

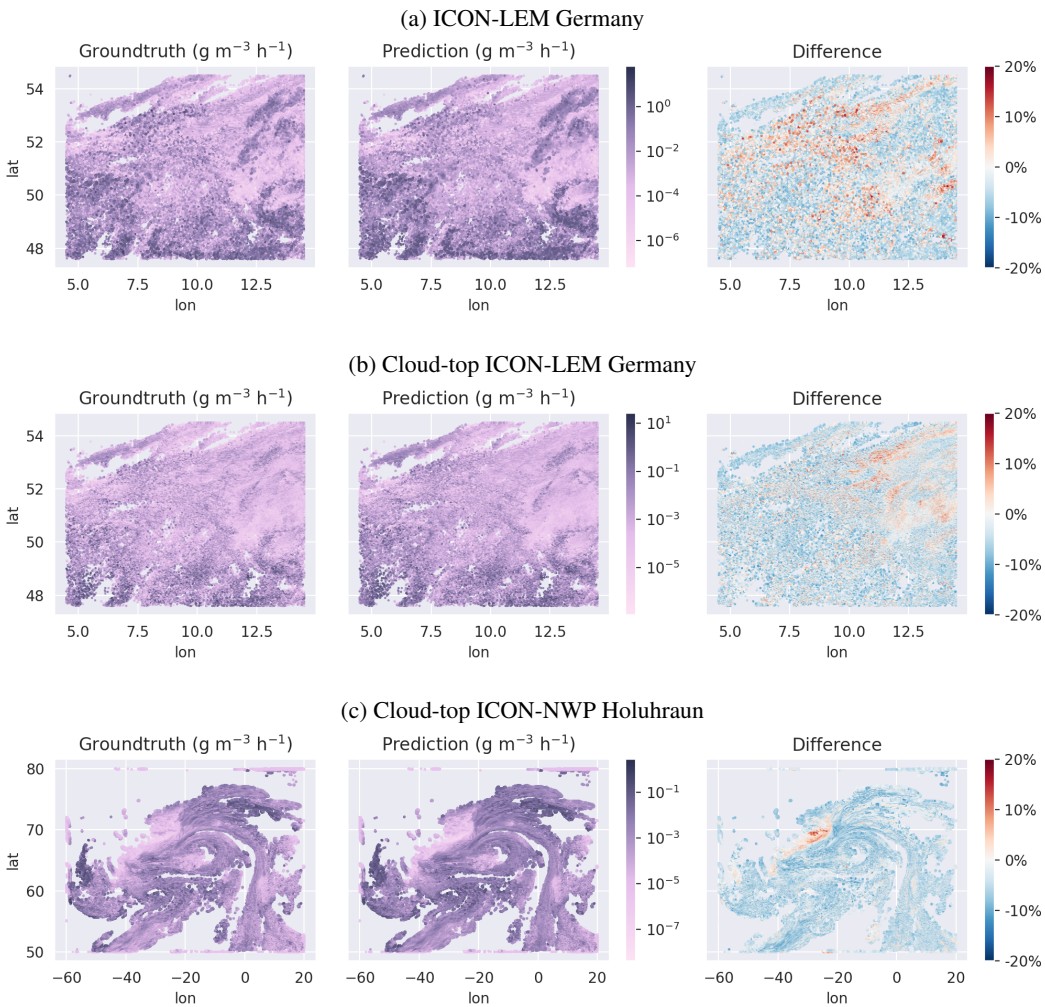

Figure E1: Visualization of the autoconversion prediction results of ICON-LEM Germany and ICON-NWP Holuhraun. The left side of the image depicts the groundtruth, while the middle side shows the prediction results obtained from the NN model. The right side displays the difference between the groundtruth and the prediction results. The top image (a) compares the groundtruth and predictions from ICON-LEM Germany at a resolution of 1 km, while the second image (b) focuses on cloud-top information only at a resolution of 1 km. The third figure (c) illustrates the comparison of comparison between groundtruth and predictions of the ICON-NWP Holuhraun data with a horizontal resolution of 2.5 km, focusing on cloud-top information only.

Based on our study, we have arrived at the finding that the liquid cloud particle size plays a crucial role in determining autoconversion rates. This impact is evident in the consistent shape of the PDF of autoconversion rates, which closely mirrors the behavior of the input.

## E.4 Uncertainty Estimation on Satellite-like and Satellite Data

Figure E4 illustrates the visualization of aleatoric and epistemic uncertainty estimations on satellite-like data over Holuhraun, while Figures E3 and E5 show the visualization of uncertainty estimations on actual satellite data over Germany and Holuhraun. These discoveries demonstrate consistency with our previous findings (refer to Section 3). It indicates that our results remain aligned and consistent across different datasets.

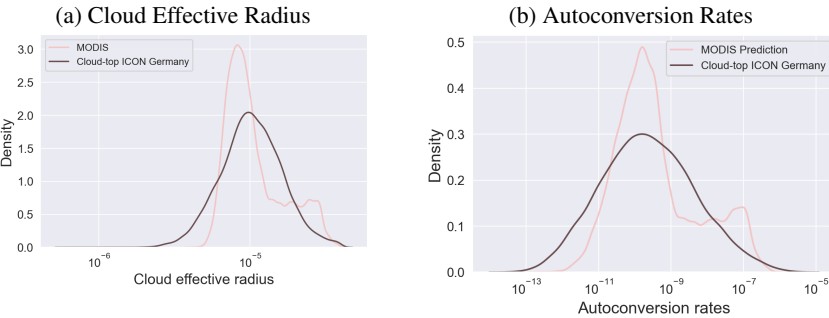

Figure E2: Probability density function of Cloud-top ICON Germany and MODIS variables: cloud effective radius and autoconversion rates.

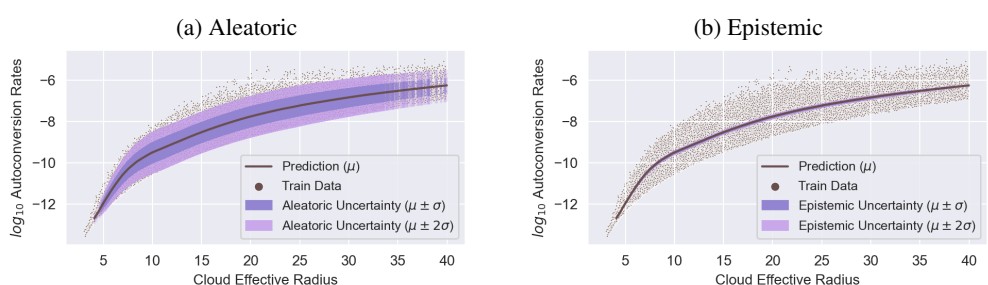

Figure E3: (a) Aleatoric and (b) epistemic uncertainty estimates of the autoconversion rates prediction $(\mathrm{kg}\,\mathrm{m}^{-3}\,\mathrm{s}^{-1})$ on satellite data (MODIS) over Germany.

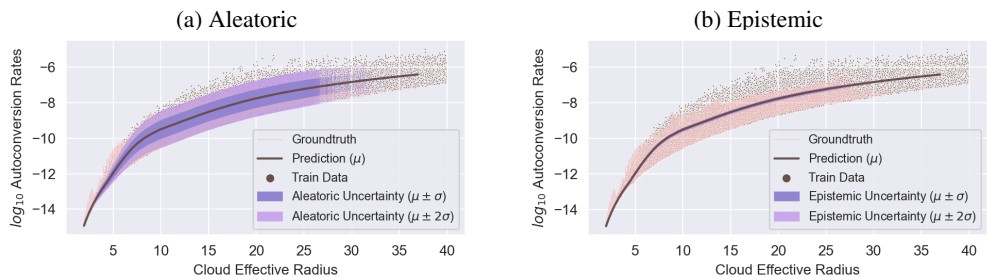

Figure E4: (a) Aleatoric and (b) epistemic uncertainty estimates of the autoconversion rates prediction $(\mathrm{kg}\,\mathrm{m}^{-3}\,\mathrm{s}^{-1})$ on atmospheric simulation data (ICON) over Holuhraun.

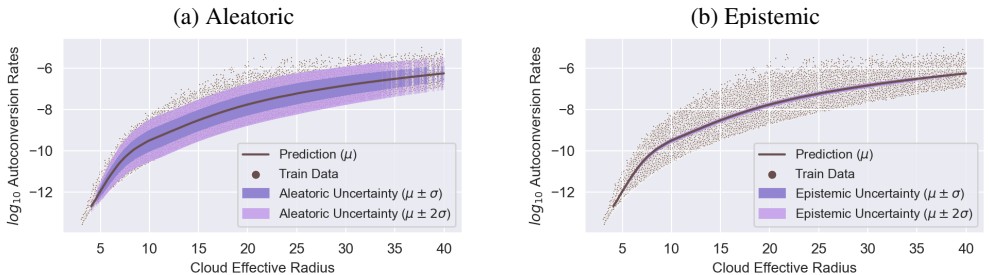

Figure E5: (a) Aleatoric and (b) epistemic uncertainty estimates of the autoconversion rates prediction $(\mathrm{kg}\,\mathrm{m}^{-3}\,\mathrm{s}^{-1})$ on satellite data (MODIS) over Holuhraun.

