# OpenReview forum: "Unleashing the Autoconversion Rates Forecasting: Evidential Regression from Satellite Data"
_NeurIPS.cc/2023/Workshop/AI4Science — NeurIPS2023-AI4Science Poster_

### Official Review · Reviewer_jBfw · 2023-10-23
**Neural networks applied to scalable atmospheric forecasting from simulations and satellite data with predictive uncertainty quantification**

**Rating:** 7
**Confidence:** 4

**Review:**

In this paper, ML and especially DNN/NN approaches are applied to forecasting of droplets formation process by means of auto-conversion rates from simulated and satellite data to effectively approximate and scale the high-computation simulations. Furthermore, evidential regression approach is utilised to estimate the predictive uncertainty of the model. Promising result with 3 ICON simulation datasets and satellite observations from MODIS are shown. As an application paper, it shown interesting utilisation of AI/ML in environmental and earth sciences combining existing methods, is well-motivated, and clearly structured. There are several experimental results to support the modelling choices. However, the experiments could be more versatile. For example, detailed analysis of computing time and computing costs between ML models and simulations are missing as well as comparison to previous work in the area. Also, there could be a baseline to compare the usefulness  of uncertainty estimation with the evidential model.

Pros
- Real scientific problem applying ML/NNs and supporting simulations
- Real-world datasets
- Considering model's predictive uncertainty
- Promising forecasting results

Cons
- No detailed comparison of computing times / costs (i.e., ML vs. simulations)
- No Comparison of uncertainty estimation to any baseline (appendix has baseline models for accuracy, though)
- Limited comparison to previous work, e.g., Novitasari et al. 2021.

Minor typos and modifications
- Figure 2 (caption text): ... shows the prediction results obtained from the GP model. -> ... shows the prediction results obtained from the NN model. (?)
- References: add all information to references (e.g., Arxiv id (if not peer-reviewed yet)

---

### Meta-Review · Area_Chair_1pAT · 2023-10-27

**Recommendation:** Accept (Poster)
**Confidence:** 4

**Metareview:**

The paper proposes the new neural network that provides the prediction of autoconversion rates with uncertainties under consideration. I believe that this paper will provide more insight to the weather forecasting community. I hope the author can address the concern, especially adding more works in the introduction, such as other probabilistic forecasting model: https://arxiv.org/abs/2307.10422